# Impact of study design, contamination, and data characteristics on results and interpretation of microbiome studies

Jose Agudelo,[1] Aaron W. Miller[1,2]

**ABSTRACT** Advances in high-throughput molecular techniques have enabled microbiome studies in low-biomass environments, which pose unique challenges due to contamination risks. While best-practice guidelines can reduce contamination by over 90%, the impact of residual contamination and data set variability on statistical outcomes remains understudied. Here, we quantitatively assessed how study design factors influence microbiome analyses using simulated and real-world data sets. Alpha diversity was affected by sample number and community dissimilarity, but not by the number of unique taxa. Beta diversity was influenced primarily by unique taxa and group dissimilarity, with a marginal effect of sample number. The number of differentially abundant taxa depended on the number of unique taxa but was also influenced by sample number, depending on thealgorithm. Notably, contamination had a marginal impact on weighted beta diversity but altered the number of differentially abundant taxa when at least 10 contaminants were present, with a greater effect as contamination increased. Findings closely mirrored results from seven real-world low-biomass data sets. Overall, group dissimilarity and the number of unique taxa were the primary drivers of statistical outcomes. The DESeq2 algorithm outperformed ANCOM-BC when exposed to stochastically distributed contamination, but algorithms were equivocal under contamination weighted toward one group. In all cases, the rate of false positives in differential abundance analyses was <15%. Importantly, in both simulated and real-world data, contamination rarely impacts whether microbiome differences were detected but did affect the number of differentially abundant taxa. Thus, when validated protocols with internal negative controls are used, residual contamination minimally impacts statistical outcomes. Alpha diversity was affected by sample number and community dissimilarity, but not by the number of unique taxa. Beta diversity was influenced primarily by unique taxa and group dissimilarity, with a marginal effect of sample number. The number of differentially abundant taxa depended on the number of unique taxa but was also influenced by sample number, depending on the algorithm. Notably, contamination had a marginal impact on weighted beta diversity but altered the number of differentially abundant taxa when at least 10 contaminants were present, with a greater effect as contamination increased. Findings closely mirrored results from seven real-world low-biomass data sets. Overall, group dissimilarity and the number of unique taxa were the primary drivers of statistical outcomes. The DESeq2 algorithm outperformed ANCOM-BC when exposed to stochastically distributed contamination, but algorithms were equivocal under contamination weighted toward one group. In all cases, the rate of false positives in differential abundance analyses was <15%. Importantly, in both simulated and real-world data, contamination rarely whether microbiome differences were detected but did affect the number of differentially abundant taxa. Thus, when validated protocols with internal negative controls are used, residual contamination minimally impacts statistical outcomes.

Address correspondence to Aaron W. Miller, millera25@ccf.org.

The authors declare no conflict of interest.

**IMPORTANCE** Microbiome studies in low-biomass environments face challenges due to contamination. However, even after implementing strict contamination prevention, control, and analysis measures, the impact of residual contamination on the validity of statistical outcomes in such studies remains a topic of ongoing discussion. Our analyses reveal that key drivers of microbiome study outcomes are group dissimilarity and the number of unique taxa, while contamination has minimal impact on statistical outcomes, primarily limited to the number of differentially abundant taxa detected. A common approach to contamination control involves removing taxa based on published contaminant lists. However, our analysis shows that these lists are highly inconsistent across studies, limiting reliability. Instead, our results support the use of internal negative controls as the most robust means of identifying and mitigating contamination. Collectively, data show that low-biomass microbiome studies have reduced power to detect differences between groups. However, when differences are observed, they are unlikely to be contamination-driven. By prioritizing validated protocols that prevent, assess, and eliminate contaminants through the use of internal negative controls, researchers can minimize the impact of contamination and improve the reliability of results.

**KEYWORDS** microbiome, low microbial biomass, simulated data, real-world data, contamination, study design

With the advancement of highly sensitive, high-throughput molecular and culture-based analytical tools, investigators have increasingly focused attention on environments with low-diversity, low-density microbial communities, many of which were previously thought to be sterile (1–4). However, these kinds of environments present unique challenges, not often considered in high-density environments such as the mammalian gut or soil. These challenges include mis-annotation of host DNA as microbial (5, 6) due to the high ratio of host to bacterial DNA, which can lead to bias in amplicon-based sequences (7), stochastic distributions of microorganisms (8–10), statistical biases driven by low diversity and stochasticity (11, 12), as well as the detection of residual microbial DNA vs viable organisms (13, 14). However, perhaps the concern that has drawn the most attention in published literature is the role that environmental contaminants play in biasing downstream results. Sources of contamination can be of two types: external sources, such as laboratory reagents, air, or other environmental sources, and cross-contamination between samples (15–19). Uncontrolled contamination has controversially led to the questioning of the discovery of a microbiome in environments previously considered sterile, such as the placenta (20–22), brain (5), or tumors (6). Given the recognized issues and challenges in dealing with contamination in low microbial biomass studies, there have been a number of best practice guidelines published that have focused on preventing, assessing, and bioinformatically removing contamination (15, 16, 19, 23–25). Following these recommendations, contamination can be reduced in downstream results by over 90% (16, 19, 26).

Separating true from false signals is a critical aspect of any scientific field of study that often requires subjective decisions based on factors such as signal intensity, potential relevance of the signal, or expectations on significant vs non-significant signals (27–32). Trying to attain 0% false signals/discoveries in research, which requires perfectly balancing subjective signal detection criteria against multiple potential sources of signal bias, is often difficult or impossible to achieve (33). Additionally, setting unreasonable expectations(33) with the available technology and potential ethical considerations can hinder progress in the field. With more conservative signal filters, there is a decreased risk of obtaining false positives, but this necessarily comes with an increased risk of obtaining false negatives. In contrast, with more liberal signal filters, there is an increased risk of obtaining false positives, with a decreased risk of false negatives (Fig. 1). For microbiome studies, in environments where the ratio of real signals vs technical artifacts or contamination is high, such as the mammalian gut or soil, the risk of obtaining false positives

due to contamination is low and in fact is not typically considered (16). However, in low microbial biomass samples, the ratio of real to artifact/contaminant signal is much lower, increasing the potential of obtaining either false positives (19) or negatives (Fig. 1).

With the considerable focus on contaminant control and filtering in the literature, there has been a push for more and more conservative filters for contaminants, which decreases the risk of false positives but comes at the expense of false negatives. Recommended criteria often include using reagent controls to assess and eliminate contaminants, defined at the amplicon sequence variant level, directly introduced into samples, as well as to remove bacteria from published lists of common kit contaminants, typically defined at the genus level (34). However, many such "kitome" lists have been published, that are often quite different from each other (16, 17, 19, 25, 35), suggesting batch-specific or lab-specific contaminants are much more relevant than common contaminants. Additionally, these lists commonly include species that have been recurrently identified as residents of low microbial biomass niches. For example, the urinary tract microbiome, or urobiome (36), is a low microbial biomass environment where multiple studies have reported the presence of taxa such as *E. coli*, *Enterobacter*, *Streptococcus*, *Acinetobacter*, *Bacillus*, and *Corynebacterium* (37–39). However, these taxa are all present, at the genus level, on some common kit contaminant lists (35). Therefore, decontaminating a low-biomass data set, such as the urobiome, using both lab-specific reagent negatives and published lists of common contaminants in laboratory reagents can be overly conservative and will necessarily lead to a significant number of false negatives (40).

Despite the considerable literature on the impact of contamination on low-biomass microbiome studies, quantitative assessments of contamination or other metrics of study design or systems on the statistical outcomes of microbiome studies are scarce. This represents a critical gap in the literature, and addressing it would help optimize approaches for managing contamination concerns and other study design considerations. As such, the objective of the current study was to quantitatively assess the impact of study and sample characteristics—such as sample number, number of taxa, microbiome dissimilarity of experimental groups, and contamination—on ecological metrics, that include alpha/beta diversity and differential abundance analyses, using both simulated and real-world data. Results indicate that the number of unique taxa and dissimilarity between experimental groups were the primary drivers of statistical outcomes, while contamination primarily impacted the number of differentially abundant taxa detected between groups. Data suggest that low microbial biomass studies have a reduced potential to detect differences between experimental groups compared to high microbial biomass environments, but if a difference is detected, results are unlikely to be driven by residual contamination. As such, if investigators follow established protocols to prevent, assess, and eliminate contaminants, which include internal negative controls, then the contribution of residual contaminants to differentially abundant taxa would be minimal. These taxa are those proposed to impact host phenotype and must be experimentally validated. Collectively, based on the results of this study, knowledge-based approaches based on the use of published lists of reagent contaminants can potentially remove true positives present in a given sample. Therefore, our data suggest that relying solely on internal negative controls that drastically reduce contaminants while preserving truly present bacteria produces more accurate microbial profiles in low microbial biomass communities.

## RESULTS

### Generation of simulated data

Simulated data were generated using the HeritSeq package (41) in R statistical software. A total of 120 unique data sets were produced by permuting the sample size (defined by the vec.num.rep parameter, with values of 10, 30, 60, 120, or 240) across two arbitrarily defined experimental groups (S1 and S2). The maximum number of taxa in any one sample (defined by the alphas parameter with values of 10, 100, 1000, 2000, or 5000),

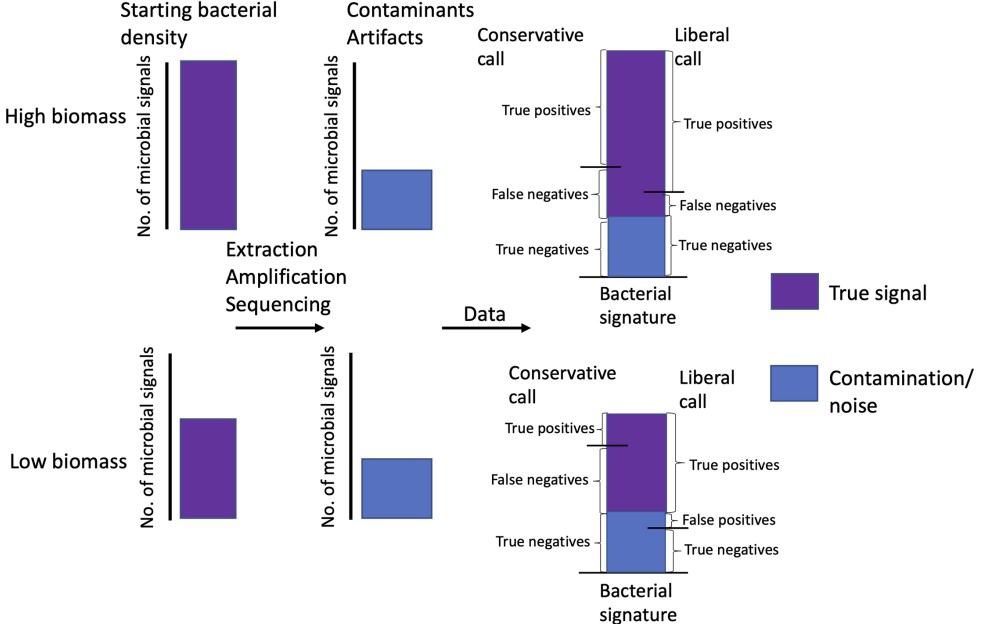

**FIG 1** Schematic of workflow for microbiome studies as it relates to signal acquisition. Schematic shows signals originating from starting sample (left), to those acquired during processing (middle), and the total amount of signal seen by investigators (right). Examples of high (top) and low (bottom) microbial biomass samples are shown. On the right, the relative impact of more conservative or more liberal quality control filters is shown, relative to the generation of true/false positives/negatives.

and the dissimilarity in microbiome composition between groups (defined by the sigma2s parameter, using the values 0.1, 1, 10, 50, and 100) were varied systematically. The code used to generate data sets is in File S1. Examples of statistical outcomes between groups S1 and S2 with values at the low, middle, and high ends of each parameter are shown in Fig. S1a through f.

## Impact of experimental and data characteristics on statistical outcomes

To evaluate the impact of experimental and data characteristics on statistical outcomes, we calculated an unweighted alpha diversity as Margalef's species richness, as well as the weighted Simpson index. Beta diversity was calculated as a weighted Bray-Curtis dissimilarity matrix and an unweighted binomial dissimilarity matrix. Finally, differential abundance analyses were performed using the DESeq2 and ANCOM-BC algorithms for all 120 simulated data sets. We found that sample number had a marginal impact on unweighted alpha and weighted beta diversity metrics, while having a considerable impact on the weighted Simpson's index (Fig. 2A and B). However, sample number did not impact the number of differentially abundant taxa detected using the DESeq2 algorithm, but marginally influenced the number of taxa detected with ANCOM-BC (Fig. 2C). In contrast to sample number, the maximum number of taxa in a sample did not impact the statistical outcomes for alpha diversity, but had a strong impact on beta diversity statistical outcomes and the number of differentially abundant taxa detected (Fig. 3A through C). Finally, the defined dissimilarity in microbiome composition had a strong impact on alpha diversity statistical outcomes, as well as the weighted beta diversity, while impacting the number of differentially abundant taxa using the ANCOM-BC but not DESeq2 algorithms (Fig. 4A through C).

## Impact of contamination on statistical outcomes

To evaluate the potential impact of contamination on statistical outcomes, we simulated two different manifestations of contamination. First, contamination was simulated where it was more heavily biased in one experimental group compared to the other, as would

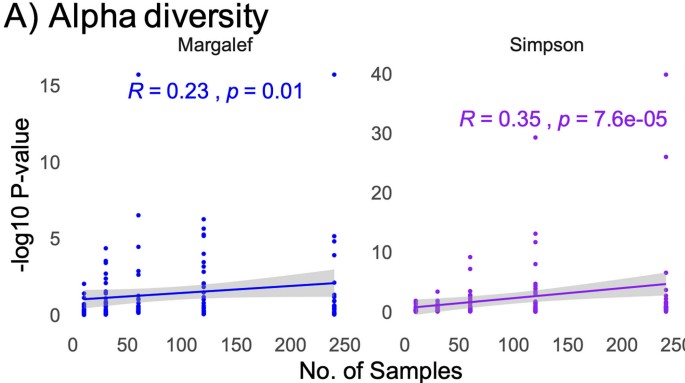

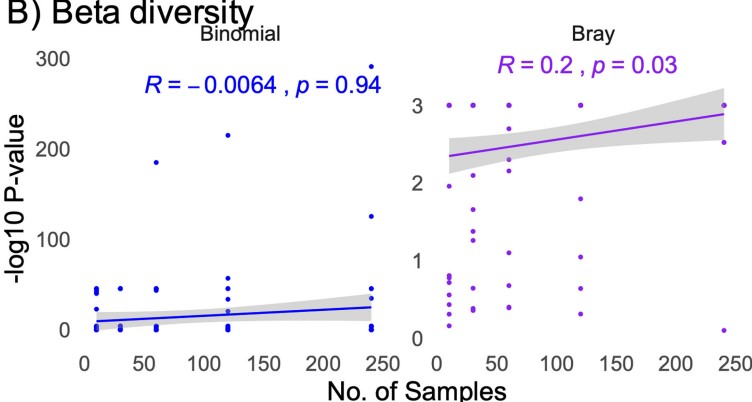

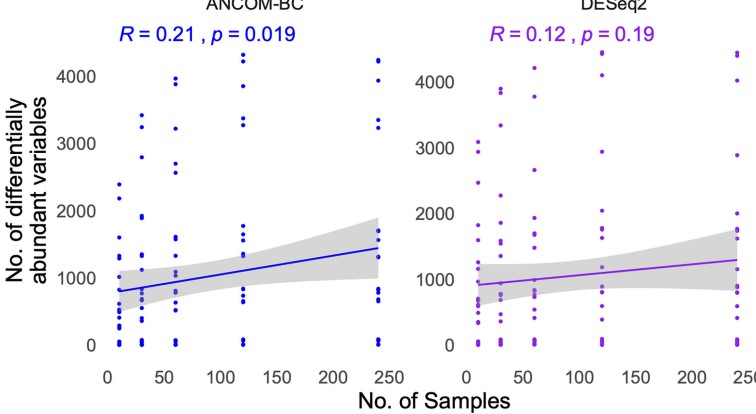

**FIG 2** Correlations between statistical outcomes and sample number. (A, B) Pearson correlations between the −log10 transformed *P* values between simulated groups (S1 and S2) and sample number for alpha diversity (A) and beta diversity (B). (C) Pearson correlation between the number of significant, differentially abundant taxa between groups and sample number. Correlation and *P* values are shown on the chart.

be seen in case:control studies of infectious disease. Secondly, we simulated a universal, stochastic distribution of contaminants, as would be seen in case:control studies of non-infectious disease. The code to generate weighted and unweighted contamination is provided in Files S2 and S3, respectively. For all 120 data sets, contamination was simulated at different levels with the maximum number of contaminants distributed as follows: 0, 1, 5, 10, or 100 contaminants (600 data sets in total).

## A) Alpha diversity

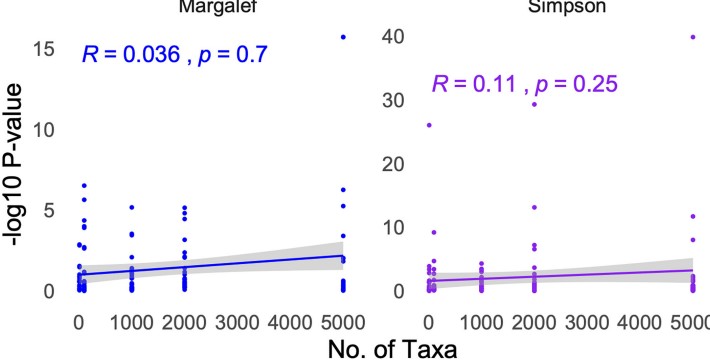

## B) Beta diversity

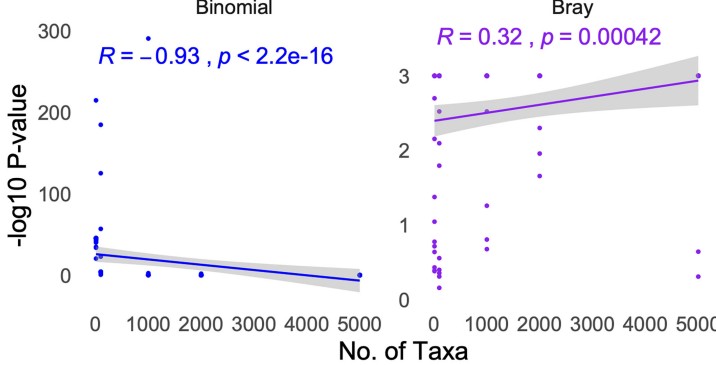

## C) Differential abundance

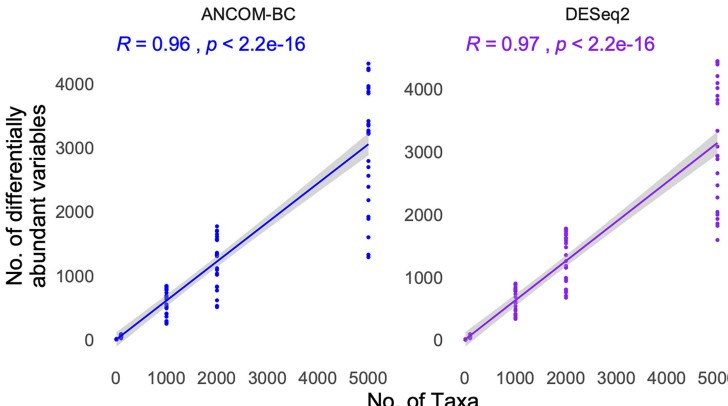

**FIG 3** Correlations between statistical outcomes and the maximum number of taxa seen in any one sample. (A, B) Pearson correlations between the −log10 transformed $P$ values between simulated groups (S1 and S2) and taxa number for alpha diversity (A) and beta diversity (B). (C) Pearson correlation between the number of significant, differentially abundant taxa between groups and sample number. Correlation and $P$ values are shown on the chart.

For the unweighted contamination, Pearson correlations to compare the number of contaminants to the fold-change in $P$ value between the original data set (0 contaminants) and those with contaminants added (1–100 contaminants) revealed that the number of contaminants in the data only had a marginal impact on unweighted beta diversity (where five contaminants led to a twofold change in $P$ value), for the diversity metrics (Fig. 5A and B). In contrast, there was a strong impact of contaminant number on the number of differentially abundant taxa identified (Fig. 5C). For ANCOM-BC, less than one contaminant changed the number of differentially abundant taxa detected by two,

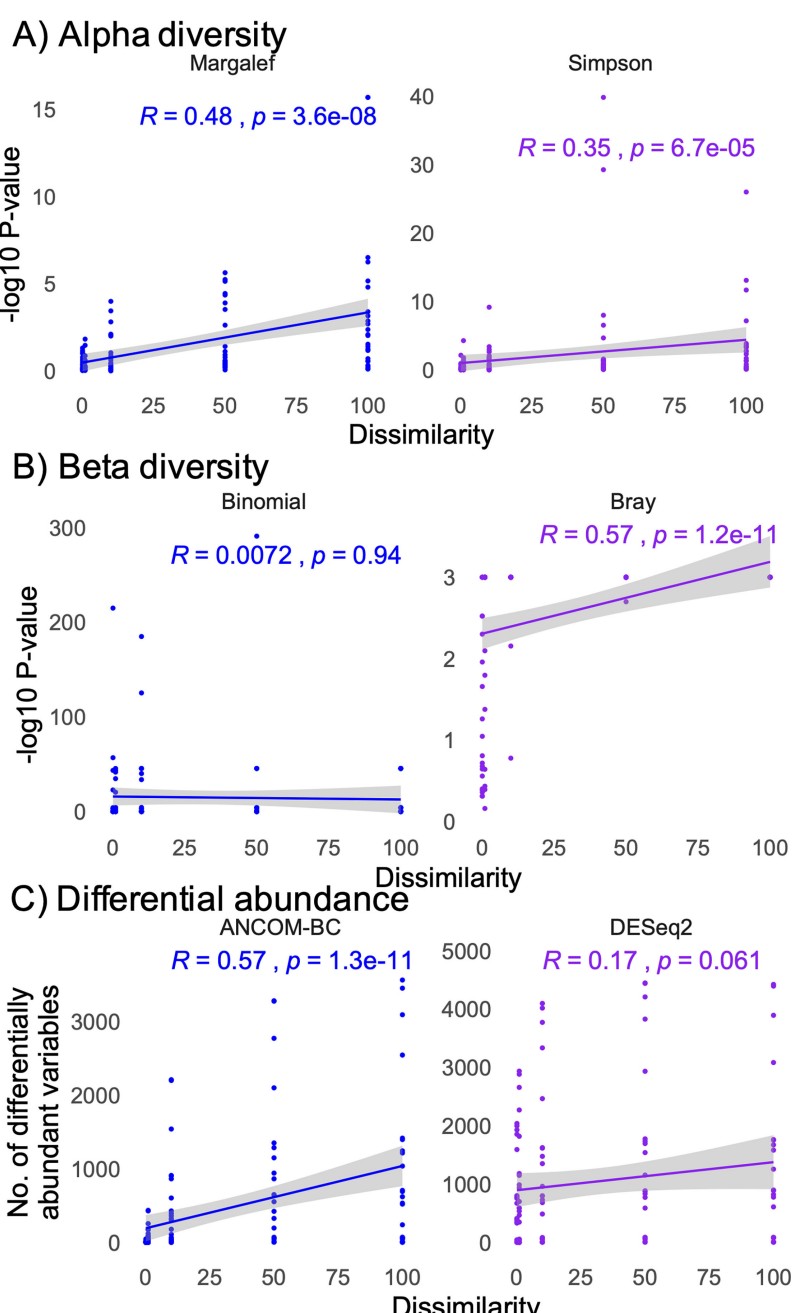

**FIG 4** Correlations between statistical outcomes and the dissimilarity between simulated groups (S1 and S2). (A, B) Pearson correlations between the −log10 transformed *P* values between simulated groups (S1 and S2) and microbiome dissimilarity for alpha diversity (A) and beta diversity (B). (C) Pearson correlation between the number of significant, differentially abundant taxa between groups and microbiome dissimilarity. Correlation and *P* values are shown on the chart.

whereas four contaminants were required to change the number of differentially abundant taxa by two.

In weighted contamination simulations, contamination did not impact alpha or beta diversity metrics (Fig. 6A and B). Weighted contamination did impact the number of differentially abundant taxa detected with DESeq2, but not ANCOM-BC. Less than one contaminant was enough to see a change of two significant hits with DESeq2.

To determine how contamination influenced the number of differentially abundant taxa, we quantified the proportion of significant hits that were contaminants, borderline

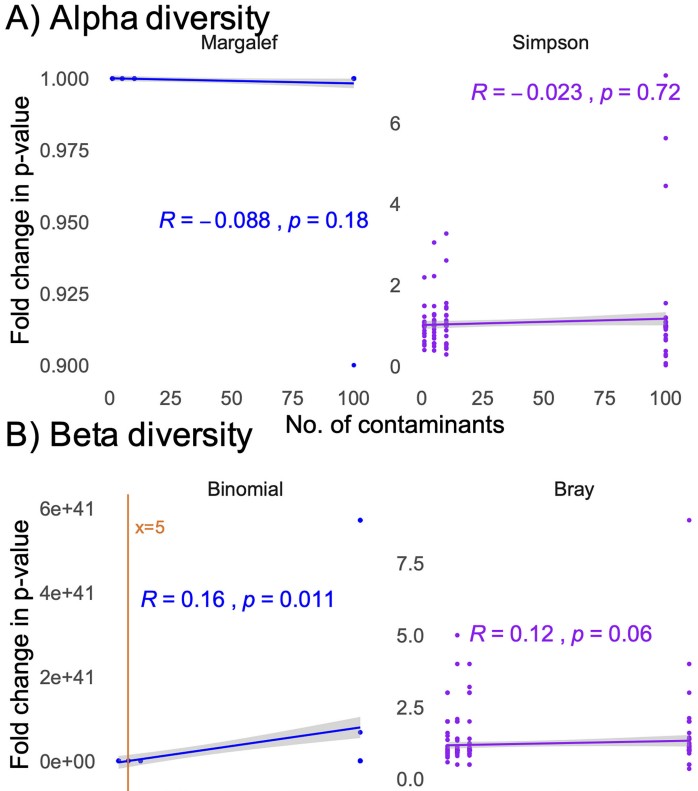

## A) Alpha diversity

## B) Beta diversity

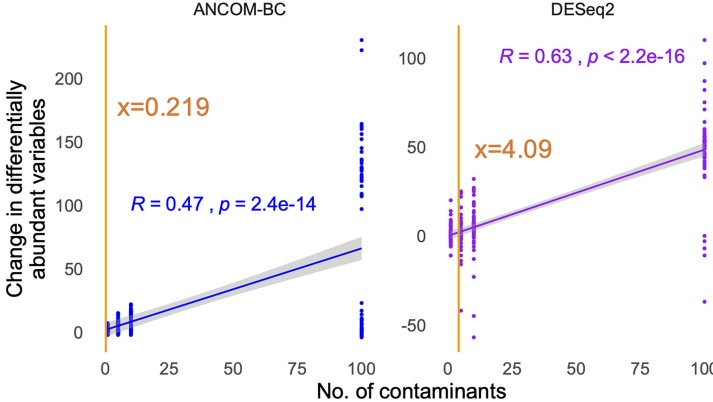

## C) Differential abundance

**FIG 5** Impact of stochastically distributed contamination on simulated data, reflected as the fold-change in *P* values (A, B) or the change in the number of differentially abundant taxa (C) between original data and data with added contaminants. Shown are impacts on alpha diversity (A), beta diversity (B), and differential abundance (C) analyses. In analyses with a significant correlation, the orange line represents the calculated number of contaminants required to see a twofold change in *P* value (beta diversity) or a difference of two differentially abundant variables (differential abundance analyses).

significant (false discovery rate-corrected *P* value [FDR] between 0.05 and 0.1 in the uncontaminated data set), or random (FDR > 0.1 in the uncontaminated data set). With unweighted contamination, false positives were primarily generated by moving random taxa to significance (Fig. S2), with the ANCOM-BC exhibiting greater shifts in differentially

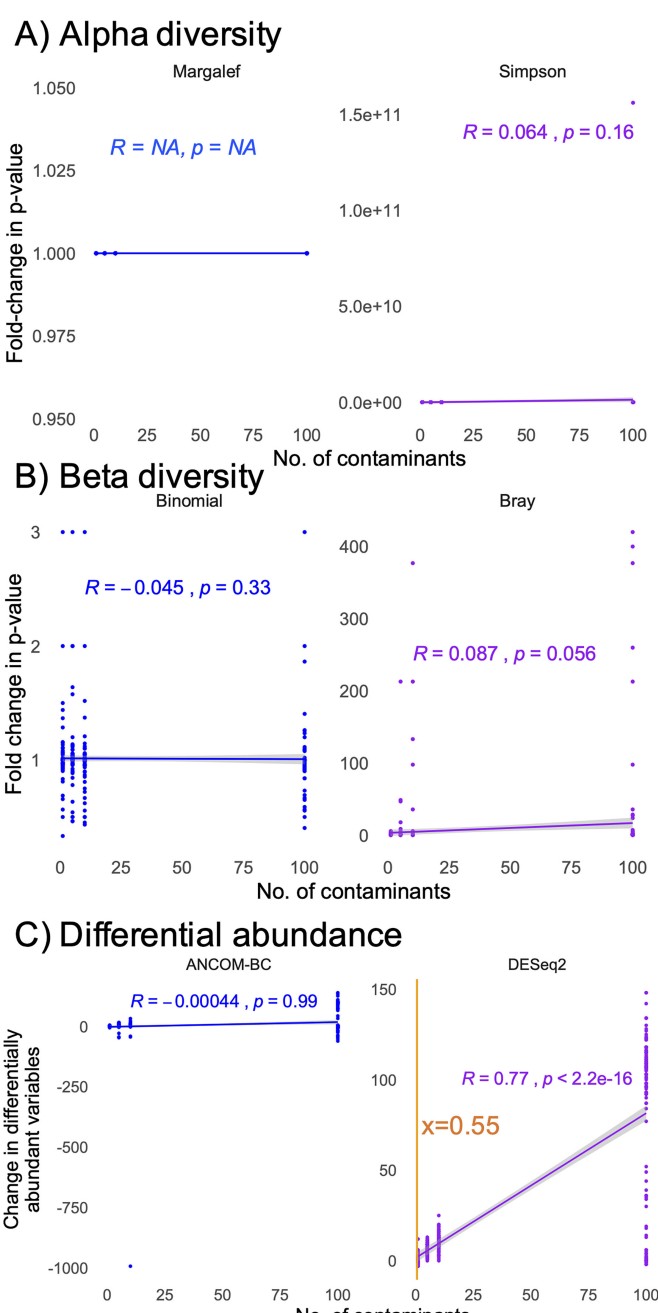

**FIG 6** Impact of stochastically distributed contamination, weighted more heavily on one group than the other, on simulated data, reflected as the fold-change in *P* values (A, B) or the change in the number of differentially abundant taxa (C) between original data and data with added contaminants. Shown are impacts on alpha diversity (A), beta diversity (B), and differential abundance (C) analyses. In analyses with a significant correlation, the orange line represents the calculated number of contaminants required to see a difference of two differentially abundant variables (differential abundance analyses).

abundant taxa than DESeq2. With weighted contamination, the two algorithms performed similarly, with greater weight toward false positives coming from the contaminants themselves (Fig. S2). In all cases, the number of false-positive results was <15% of the total number of significant features.

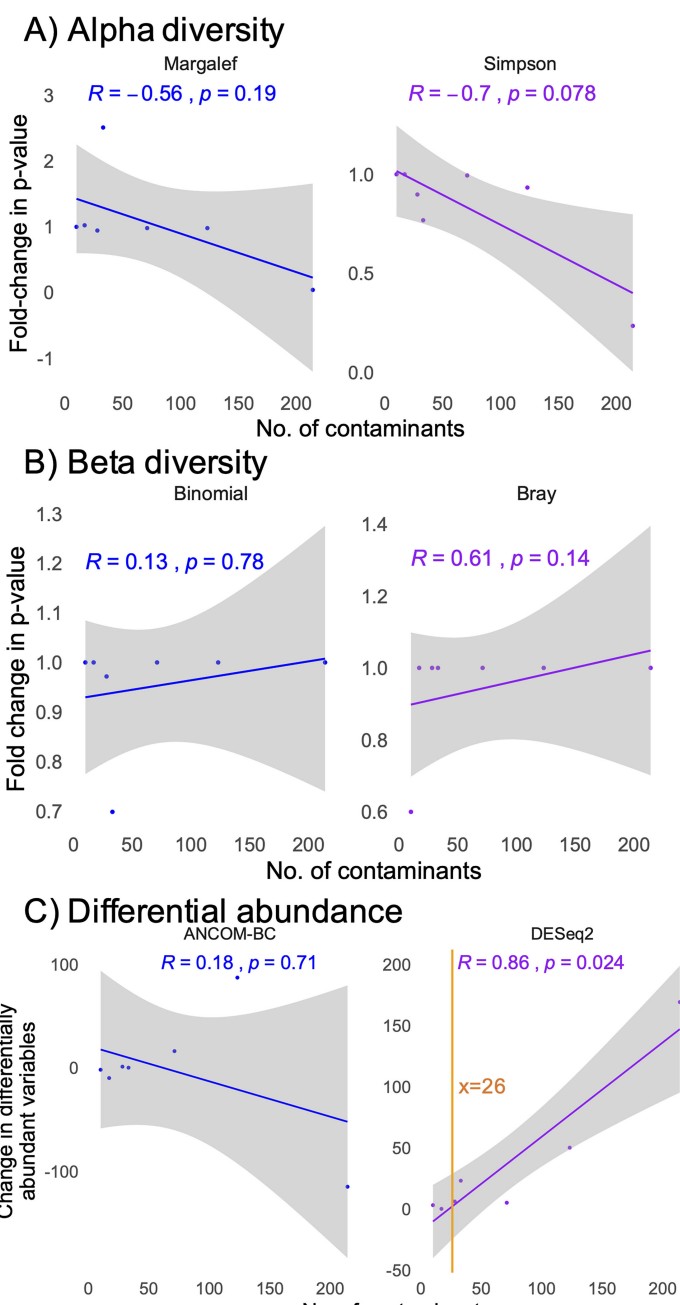

**FIG 7** Impact of contamination on real-world data sets, reflected as the fold-change in *P* values (A, B) or the change in the number of differentially abundant taxa (C) between original data and data with contaminants removed. Shown are impacts on alpha diversity (A), beta diversity (B), and differential abundance (C) analyses. In analyses with a significant correlation, the orange line represents the calculated number of contaminants required to see a difference of two differentially abundant variables (differential abundance analyses).

## Validating the impact of contaminants with real-world data

To validate the impact of contamination on statistical outcomes from simulated data, we utilized seven real-world data sets generated from clinical, low microbial biomass, urogenital tract specimens. These data sets included samples from midstream urine, semen, explanted urologic devices (for any reason), or bladder tumor tissue, representing a total of 688 samples (SAMN42782311-2536, SAMN44486723-959,

SAMN44403240-313, SAMN43044911-46, SAMN43036145-253, SAMN42560143-258, and SAMN43011969-2070). For real-world data, the difference in $P$ values or number of differentially abundant taxa was assessed using statistical outcomes derived from data with low quality and host reads removed, but without removing internal negative controls (laboratory reagents). This approach contrasts with the same data, where contaminants were removed using the Decontam package (42) with internal negative controls serving as the source of contamination. Similar to the simulated data for weighted contaminants (Fig. 6), the number of contaminants present in samples did not significantly correlate with alpha or beta diversity outcomes. However, it did significantly correlate with the number of differentially abundant taxa detected for DESeq2, but not ANCOM-BC (Fig. 7A through C). The number of contaminants needed to change the number of differentially abundant taxa by two was 26 (Fig. 7C).

## DISCUSSION

In the last decade, novel, high-throughput molecular and culture-based techniques have suggested that environments previously thought to be sterile actually harbor resident microbial communities (1–4). As with any new area of investigation, concerns have been raised about the validity of conclusions drawn from environments proposed as harboring no microorganisms or a low microbial biomass community. Chief among these concerns is the potential for contamination, as a major determinant of the data generated from these environments (15–17, 19). As a result, a number of studies have focused on the "kitome"—a list of common contaminants identified when sequencing the reagents used in sample processing and sequencing—with the suggestion that these contaminant lists can be used to decontaminate studies of low-biomass environments (16, 17, 19, 21, 25, 26, 35). However, published lists of common reagent contaminants are quite different from each other. Across seven published lists (16, 17, 19, 21, 25, 26, 35), 429 genera have been identified as common contaminants. Of these, 351 genera only show up in one list, while 53 only show up in two lists. Only 18 of the 429 genera appeared in >50% of published lists, with none being present in all lists (Table S1). More importantly, these lists, which typically report at the genus level, include many bacteria, such as *Escherichia*, *Staphylococcus*, or others, that have been recurrently identified in strictly controlled studies of low microbial biomass niches, such as the urinary tract (37–39). Based on these findings, the use of these lists could not be a valid approach to decontaminating data. In contrast, using established recommendations that include the use of rigorous internal negative controls, at every step of the process, combined with statistical approaches to remove contaminants, has been shown to reduce contamination by 90% while preserving data from bacteria truly present in the study environment (18, 23, 24).

While there has been considerable literature dedicated to the impact of contamination or other technical artifacts on studies of low microbial biomass, quantitative assessments of these effects on statistical outcomes are scarce. Results from our simulated data, examining 120 unique data sets with or without variable amounts of weighted or unweighted contamination, reveal that the primary drivers of statistical outcomes are the dissimilarity in microbiome composition between study populations and the number of unique taxa present. In diversity metrics, contamination (unweighted) only had a marginal impact on unweighted beta diversity. Importantly, diversity metrics determine whether or not there is a difference in the microbiome between two populations. The primary impact of contamination was on the number of differentially abundant taxa present between populations. Our data indicate that while contamination can shift which taxa are significantly different between groups, the total number of false positives is <15% under both weighted and unweighted simulations.

Collectively, the data show that microbiome differences in low-biomass environments between populations are unlikely to be driven by contamination if standardized approaches are implemented (16). However, a major goal of clinical microbiome studies is to identify modifiable microbial signatures that can be used to alter clinical outcomes toward better health. Differential abundance analysis, which detects the specific

species associated with health or disease, provides the potential targets of modification. While differential abundance analysis was the primary statistical outcome impacted by contamination, the proportion of false positives was <15% even at high levels of contamination, and identified targets require experimental validation to show that they actually influence a particular environment. By following established recommendations to prevent, assess, and eliminate contamination, the influence of contaminants on differential abundance analyses would be minimized, thus limiting the detection of false-positive signatures that must be validated. We note that if false-positive microbial targets are incorrectly attributed to a disease phenotype and lead to novel therapeutics, then this would lead to a failed therapeutic in clinical trials. In contrast, false-negative microbial signatures incorrectly not attributed to a disease phenotype would prevent the development of novel therapeutics that could lead to a real benefit for patients.

Studies of low microbial biomass environments are in their infancy. Studies into host-associated, low microbial biomass environments really started as a field less than two decades ago (43–45). The assumption of sterility, particularly in host-associated environments, is in large part driven by inadequate assays to detect bacteria and a disconnect between technical definitions of sterility, which the World Health Organization defines as the freedom from the presence of viable microorganisms (46), and clinical definitions of the term. Specifically, many clinical assays to detect bacteria in host-associated environments, developed over 50 years ago, are specifically designed to detect a high density of fast-growing, aerobic bacteria, indicative of infection (47). When assays do not produce bacterial numbers above a defined threshold, they are considered "sterile." If the bar for detection is lowered by increasing sampling effort and culture conditions, the frequency of detection is considerably higher (47), which indicates that samples considered clinically sterile are not likely to be considered sterile by the definition provided by the World Health Organization. Given the infancy of low-biomass, host-associated microbial communities, we do not know the true distribution of bacterial species in these environments, nor how they influence host physiology. To advance the field, efforts must be made to accurately assess what bacteria are truly present in these environments while balancing false-positive signals. Results from the current study make clear that environmental contamination is batch (lab) specific and should be treated as such in order to adequately remove contaminants while preserving true signals.

The current study is not without limitations. In simulated and real-world data, we only assessed study design, study system factors, and contamination on statistical outcomes. We did not assess biases attributed to mis-annotation, statistical effects of stochasticity, residual DNA vs DNA from viable bacteria, or batch-specific contaminants.

In conclusion, we show that differences in low-biomass microbial communities between two defined communities are unlikely to be driven by false positives or technical artifacts, in a study that follows established protocols to prevent, assess, and eliminate contaminants (23, 24), for most of the ecological metrics evaluated. Based on the variability observed among published lists of common reagent contaminants and well-controlled studies which show that use of internal negative controls can effectively reduce contaminants by 90%, we recommend the latter approach when designing studies of low microbial biomass environments.

## MATERIALS AND METHODS

### Generation of simulated data

To determine how the number of variables (i.e., bacterial taxa), number of samples per group, the dissimilarity of data between populations, and contamination impact the statistical outcomes of common analyses with microbiome data, we generated simulated data using the HeritSeq package (41) in R statistical software. A total of 600 data sets were generated to robustly examine the impact of each of these factors on statistical outcomes. To do so, data sets were generated using permutations of sample number, maximum number of variables, dissimilarity, and number of contaminants. The

sample number was defined by the vec.num.rep parameter in the HeritSeq package, with sample numbers of 10, 30, 60, 120, or 240 per two arbitrarily defined experimental groups (S1 and S2). Sample numbers were varied based on estimates from published clinical microbiome studies that span small pilot studies to larger, well-powered studies. The maximum number of taxa in any one sample was defined by the alphas parameter with values of 10, 100, 1,000, 2,000, or 5,000. These values produced the maximum number of unique simulated taxa in any one sample, as some proportion of taxa would have a count of 0, similar to real-world data. The minimum and maximum values chosen here were based on past studies of very low/defined microbial communities, through high-diversity environments such as the soil or mammalian gut. The dissimilarity in microbiome composition between groups was defined by the sigma2s parameter, using the values 0.1, 1, 10, 50, and 100. These values produce varying levels of dissimilarity based on the presence/absence of simulated taxa between groups and ranged from nearly identical populations between groups to no possible overlap. The code used to generate original, uncontaminated data sets is in File S1.

Finally, to simulate contamination, we either assumed that contamination would be present as a universal, stochastic distribution (unweighted), which assumes roughly equal starting bacterial densities between groups, or that contamination would be more heavily biased in one group than the other (weighted), which assumes disparate starting bacterial densities between groups. The number of contaminants simulated ranged from 0, 1, 5, 10, or 100 contaminants. The code used to simulate weighted or unweighted contamination is provided in Files S2 and S3, respectively. Counts for each contaminant were generated through a random number generator (rnorm() in base R). Values for the random number generated ranged from 0 to $N$, meaning that there was a chance that contaminants would have a 0 value, proportional to the total potential number of contaminants present in any given sample.

## Real-world data

To validate the results of simulated data, seven real-world data sets that all included internal negative controls in the study were examined (SAMN42782311-SAMN42782536, SAMN44486723-SAMN44486959, SAMN44403240-SAMN44403313, SAMN43044911-SAMN43044946, SAMN43036145-SAMN43036253, SAMN42560143-SAMN42560258, and SAMN43011969-SAMN43012070). Data were all generated from clinical, low microbial biomass, urologic specimens that included midstream urine, semen, explanted urologic devices (for any cause), or bladder tumor tissue. Across the seven data sets, there were 688 samples in total. For real-world data, sample number, maximum number of variables, and group divergence were not manipulated. The number of contaminants was statistically determined using the Decontam package (42), based on taxa present in internal, study-specific, negative controls. Alpha/beta diversity and differential abundance analyses were conducted on decontaminated data. The effect of contamination was evaluated by comparing decontaminated data to raw data.

## Statistical analyses

Unweighted and weighted alpha diversity between generated groups S1 and S2 was calculated as Margalef's species richness and Simpson's index for all 600 data sets, followed by paired $t$-test statistical analyses (48). Similarly, beta diversity was calculated as a weighted Bray-Curtis dissimilarity matrix or unweighted binomial dissimilarity matrix, followed by a Permutational Multivariate Analysis of Variance (PERMANOVA) with 999 permutations (48, 49). Differential abundance analyses were calculated by false discovery rate-corrected DESeq2 analyses (50) or ANCOM-BC analyses (51). In all analyses, specific taxonomy was not considered, as data were simulated and the taxonomic denominations were arbitrary. To evaluate the impact of each of the parameters tested, Pearson correlations were conducted between the alpha/beta diversity $-\log_{10}(P$ values) or the number of differentially abundant taxa, compared to sample number, maximum variable number, or group divergence. The $-\log_{10}(P$ value)

metric increases with decreasing raw $P$ value, where $-\log10(1.301)$ is equal to 0.05, the threshold of significance. To assess the impact of contamination, Pearson correlations were made between the fold-change in $P$ values with and without contaminants for alpha/beta diversity and differential abundance metrics. Here, values between 0 and 1 reflect a decrease in $P$ value, whereas values >1 reflect an increase in $P$ value. Tests of normality for data distributions were conducted prior to correlations.

To determine how contamination impacted differential abundance results, the significantly different features in the contaminated data sets were categorized as contaminants, which were directly simulated, borderline significant, having an FDR between 0.05 and 0.1 in uncontaminated data sets, or random, having an FDR > 0.1 in uncontaminated data sets. The proportion of false positives was statistically compared using a two-way analysis of variance against the differentially abundant algorithm and the type of false positive (contaminant, borderline, or random).

## ACKNOWLEDGMENTS

J.A. and A.W.M. contributed equally to study design, data analysis/interpretation, and manuscript writing. Both authors read and approved the final manuscript.

## AUTHOR AFFILIATIONS

[1]Department of Cardiovascular and Metabolic Sciences, Cleveland Clinic, Cleveland, Ohio, USA

[2]Department of Urology, Glickman Urological and Kidney Institute, Cleveland Clinic, Cleveland, Ohio, USA

## AUTHOR ORCIDs

Aaron W. Miller http://orcid.org/0000-0001-8342-1449

## AUTHOR CONTRIBUTIONS

Jose Agudelo, Conceptualization, Data curation, Formal analysis, Investigation, Methodology, Writing – original draft, Writing – review and editing | Aaron W. Miller, Conceptualization, Data curation, Formal analysis, Investigation, Methodology, Project administration, Validation, Writing – original draft, Writing – review and editing

## DATA AVAILABILITY

Publicly available real-world data sets are available through the Sequence Read Archive under accession numbers SAMN42782311–SAMN42782536, SAMN44486723–SAMN44486959, SAMN44403240–SAMN44403313, SAMN43044911–SAMN43044946, SAMN43036145–SAMN43036253, SAMN42560143–SAMN42560258, and SAMN43011969–SAMN43012070. The code to generate and analyze the simulated data is provided in File S1.

## ETHICS APPROVAL

Real-world data sets used in the current study come from publicly available data and do not constitute human or animal subjects. As such, ethics approval is not required.

## ADDITIONAL FILES

The following material is available online.

Supplemental Material

**File S1 (mSystems00408-25 S0001.txt).** Code to generate and analyze simulated data.

**File S2 (mSystems00408-25 S0002.txt).** Code used to generate weighted contamination in simulated data sets.

**File S3 (mSystems00408-25 S0003.txt).** Code used to generate unweighted contamination in simulated data sets.

**Figure S1 (mSystems00408-25 S0004.pdf).** Characteristics of simulated data sets.

**Figure S2 (mSystems00408-25 S0005.pdf).** Delineation of false positives due to contamination.

**Table S1 (mSystems00408-25 S0006.txt).** Distribution of taxa in published lists of contaminants.

## Open Peer Review

**PEER REVIEW HISTORY (review-history.pdf).** An accounting of the reviewer comments and feedback.

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
