## [Reviewer comments · mSystems]

Impact of study design, contamination, and data characteristics on results and interpretation of microbiome studies

Jose Agudelo and Aaron Miller

Corresponding Author(s): Aaron Miller, Cleveland Clinic

Review Timeline:

Submission Date:	March 19, 2025
Editorial Decision:	April 17, 2025
Revision Received:	June 13, 2025
Accepted:	June 16, 2025

Editor: Jack Gilbert

Reviewer(s): The reviewers have opted to remain anonymous.

Transaction Report:

DOI: <https://doi.org/10.1128/msystems.00408-25>

Re: mSystems00408-25 (**Impact of study design, contamination, and data characteristics on results and interpretation of microbiome studies**)

Dear Dr. Aaron W. Miller:

I am vacillating between giving you the option of a reject with resubmission or a minor edits - I think you can actually address reviewer 1 'a concerns - which are the major concerns - so resubmit, with the understanding that I will be looking out for a solid edit to address those comments or a robust rebuttal.

Revision Guidelines

Sincerely,
Jack Gilbert
Editor
mSystems

Reviewer #1 (Comments for the Author):

The primary conclusion - that results in real world low biomass samples are unlikely to be contamination driven is a dangerous message to be sending in general without strong evidence. This statement can only be true only when biomass is not linked to

the outcome of interest being investigated or is not a confounding factor in the experiment. The presented experiment does not demonstrate this issue and has substantial flaws that are problematic for drawing such string conclusions. There are bold statements about alpha and beta diversity not being affected by contaminants, though only one (not widely used) measure of alpha diversity (Margalef's) and only one measure of beta diversity (Bray Curtis) are evaluated. This is not thorough enough. Additionally, only one differential abundance analysis, DeSeq2, is evaluated which has been criticized for typically identifying significant features that are at higher relative abundance (see Nearing et al Nature Communications volume 13, Article number: 342 (2022))

The addition of contaminants in the simulated dataset does not reflect the properties of real-world contaminants, many of which are differentially amplified in relation to the biomass of a sample in a real world experiment (not random as in the simulated data). Additionally, it is unlikely that the simulated data has similar properties to microbiome sequencing data.

The presented analysis and supportive figures are also difficult to interpret. Several of the figures (Fig 2 - 5) demonstrate seeming discrete data that has a continuous analysis / regression applied to it. The outcomes also seem problematic -measuring the number of differentially abundant variables; that does not indicate that contaminants are or are not problematic for a given study.

More generally, in line 118 - Using the word strain here is incorrect; this is typically done at the sequence (amplicon sequence variant or operational taxonomic unit) level. Strains are notoriously difficult to define and detect in sequence based studies, especially marker gene or amplicon sequencing.

Reviewer #2 (Comments for the Author):

This study utilizes mock (generated) and true data sets to address key concerns about microbiome profiling in low biomass host-associated samples (e.g. urine). The authors report that alpha diversity was affected by sample number and community dissimilarity; beta diversity was influenced by unique taxa and group dissimilarity, and the number of differentially abundant taxa depended on the number of unique taxa. Contaminants altered differentially abundant taxa as well. This is important and relevant work as rigorous and validated approaches are critical in low biomass samples to prevent false positives. I include a few suggestions below. Notably, providing more biological explanations or examples to help the reader interpret the results will be beneficial in helping a broader audience appreciate the value of this work:

- o Fig S1: Explain each figure in the legend. The legend currently states "Data characteristics are labelled above graphs in the order of "number of samples in the arbitrary groups (S1,S2), maximum number of taxa in any one sample, and dissimilarity in microbiome composition" Fig S1A has "10,10,0.1" above the graph - Does this mean 10 samples in each group, 10 taxa max in any sample, and 0.1 dissimilarity? Please fully explain each figure A, B, C, etc or more clearly label what the 3 numbers above each graph are as this was not intuitive.
- o Line 185 -"and a strong impact on" - should be "HAD a strong impact on"?
- o Why Margalef's species richness as opposed to other alpha diversity metrics? (Would be helpful to explain in manuscript).
- o Further explaining the results in biological terms would be helpful. For example: You state that "Sample number had only marginal impacts on the statistical outcomes for alpha and beta diversity" Fig 2a and b show increased sample number with statistically increased $-\log_{10} p$ value. What does this mean? P values get smaller as sample number increases? Is there a threshold sample number under which you cannot obtain significance? (Or if there are low sample numbers that obtain significance, what are the other qualities of this data set that lend themselves to significance?) This type information would help me interpret these data as I consider future studies. The same is true for Fig 2B, 3A/B and 4A/B
- o For Fig 5 and 6 the y axes are "fold change in p value" - is this fold change increase? Decrease? Either?
- o It seems unusual that contaminant abundances would not alter alpha diversity outcomes. (Although I understand that what you are capturing is actually p value fold change across 2 simulated groups with 600 datasets.) How should I apply this information? For example, if I am considering UTI (high biomass) vs. healthy (low biomass) urine. The healthy urine is more likely to amplify contaminant sequences. Should I interpret from your results that I don't need to worry about the "extra diversity" from contaminants in the healthy samples as it is ultimately unlikely to change the p value between the UTI/healthy group? (Explaining some of your results in terms of other example studies can help the readers more tangibly grasp the value of this work while also serving as a "gut check" on data interpretations.
- o There was a strong impact of contaminant number on the number of differentially abundant taxa. What would be very helpful to know beyond this is how the contaminant numbers shape the differential taxa outcomes. For example, at high contaminant numbers, are the contaminant taxa showing up as differentially abundant (HUGE concern), or are they just pushing "borderline" significant taxa to significance (slightly less concern)? Or are they elevating other random taxa to significance (Concerning)? Or all of the above. Highlighting a few examples within the range of your results will make this much more tangible and applicable for readers.

The authors thank the reviewers for the constructive criticism which has greatly improved the quality of the manuscript. Below, we detail the responses to each comment in purple. We label the line numbers where changes to the manuscript can be found. Line numbers refer to the tracked changes version of the manuscript.

Reviewer #1 (Comments for the Author):

Critique 1: The primary conclusion - that results in real world low biomass samples are unlikely to be contamination driven is a dangerous message to be sending in general without strong evidence. This statement can only be true only when biomass is not linked to the outcome of interest being investigated or is not a confounding factor in the experiment. The presented experiment does not demonstrate this issue and has substantial flaws that are problematic for drawing such string conclusions. There are bold statements about alpha and beta diversity not being affected by contaminants, though only one (not widely used) measure of alpha diversity (Margalef's) and only one measure of beta diversity (Bray Curtis) are evaluated. This is not thorough enough. Additionally, only one differential abundance analysis, DeSeq2, is evaluated which has been criticized for typically identifying significant features that are at higher relative abundance (see Nearing et al Nature Communications volume 13, Article number: 342 (2022))

Response: We thank you for this comment. Indeed, the issue of contamination is nuanced. Our paper advocates for investigators to follow well-established, consensus-based, recommendations for studies involving low microbial biomass, including robust and specific measures to prevent, assess, and eliminate contamination. After applying these measures, our data strongly support the conclusion that any residual contamination or technical artifacts are unlikely to impact statistical outcomes. This conclusion is based on 600 simulated datasets that were systematically varied on multiple data characteristics, including conclusion, and results validated through multiple realworld datasets. We have further bolstered the advocacy of using recommended practices to prevent, assess, and eliminate contamination in the conclusion (among other places originally stated), as follows: "In conclusion, we show that differences in low biomass microbial communities between two defined communities are unlikely to be driven by false positives or technical artifacts, in a study that follows established protocols to prevent, assess, and eliminate contaminants^{23,24}, for most of the ecological metrics evaluated." (lines 364-367).

Given the importance of the results and the perception of flaws, we have completely recalculated all metrics in the manuscript and have now included both unweighted and weighted versions of alpha and beta diversity, along with two different metrics of differential abundance (DESeq2 and ANCOM-BC). Additionally, we completely re-did the contamination simulation in two different ways. First, we revised and updated the stochastic distribution of contaminants weighted equally across the two study groups. The updated code to simulate the equally weighted contamination is provided in supplemental file 2. Second, to address reviewer concerns, we also repeated the contaminant simulation and subsequent analyses after weighting contaminants more heavily for one group than the other. The code to simulate weighted contamination is provided in supplemental file 3. Figures 2-7 have been updated accordingly. This includes a new figure 6 (the old figure 6 is now figure 7) that includes analyses on the weighted contamination.

We note that after re-doing all of the analyses and contaminant simulations, while there are some minor differences between weighted and unweighted diversity metrics, the conclusions that residual contamination only really impacts the number of differentially abundant taxa remains the same. However, interestingly, there are differences between differential abundance algorithms in direction (increasing false positives or false negatives) and weighting contamination actually eliminates the change in differentially abundant features for ANCOM-BC, while DESeq2 is still affected.

In addition to updating all figures, we have also updated the methods, results, conclusions, and figure legends as needed. Both tracked and clean versions of the manuscript is provided for convenience.

Finally, we try to make the case, strongly backed by the data presented here, that investigators should use experiment-specific negative controls to assess and eliminate contaminants, rather than "lists of common contaminants" which we show are ambiguous and study specific, leading to the generation of a

significant number of false negatives. By using experiment-specific negatives, investigators come much closer to a microbiome profile that resembles what is biologically there and physiologically relevant, rather than incorrectly removing taxa that are likely impacting the biology in their study system. We have made this clearer (**lines 73-78, 399-403**).

Critique 2: The addition of contaminants in the simulated dataset does not reflect the properties of real-world contaminants, many of which are differentially amplified in relation to the biomass of a sample in a real world experiment (not random as in the simulated data). Additionally, it is unlikely that the simulated data has similar properties to microbiome sequencing data.

Response: Thank you. We agree that the relation starting biomass/contaminants can impact the level of differentially amplified contaminants in downstream data. In order to address this concern, we have now added a simulation of weighted contamination that represents differential amplification of taxa. In addition, in the original version, all metrics were also calculated in real-world data, which have been updated here. These analyses, using real-world, low biomass datasets produced very similar results and conclusions about contamination, thereby validating the likeness of our simulated data to real-world datasets. As stated, figures and text were updated as needed.

Critique 3: The presented analysis and supportive figures are also difficult to interpret. Several of the figures (Fig 2 - 5) demonstrate seeming discrete data that has a continuous analysis / regression applied to it. The outcomes also seem problematic -measuring the number of differentially abundant variables; that does not indicate that contaminants are or are not problematic for a given study. More generally, in line 118 - Using the word strain here is incorrect; this is typically done at the sequence (amplicon sequence variant or operational taxonomic unit) level. Strains are notoriously difficult to define and detect in sequence based studies, especially marker gene or amplicon sequencing.

Response: Thank you. For the discrete values (i.e., number of differentially abundant variables), tests of normality were performed to ensure validity of analyses. We have now added this information (**Lines 444-446**). For the number of differentially abundant variables, statistical analysis was used to determine whether different experimental metrics, including contamination, impacted the number of differentially abundant taxa detected. Our data make clearly show that contamination can specifically increase this number, highlighting the need for caution in interpretation and for the importance of follow-up studies for validation (**lines 52-54, 322-326**). We have also added new analyses to determine exactly how contamination impacts differential abundance results by calculating the proportion of significant hits that are contaminants, borderline significant, or random (**Figure S2**), which offers more granularity on the impact of contamination.

For line 118, we changed “strain” to “ASV”.

Reviewer #2 (Comments for the Author):

Critique 1: This study utilizes mock (generated) and true data sets to address key concerns about microbiome profiling in low biomass host-associated samples (e.g. urine). The authors report that alpha diversity was affected by sample number and community dissimilarity; beta diversity was influenced by unique taxa and group dissimilarity, and the number of differentially abundant taxa depended on the number of unique taxa. Contaminants altered differentially abundant taxa as well. This is important and relevant work as rigorous and validated approaches are critical in low biomass samples to prevent false positives. I include a few suggestions below. Notably, providing more biological explanations or examples to help the reader interpret the results will be beneficial in helping a broader audience appreciate the value of this work:

Response: Thank you for the positive remarks. Below we detail revisions as requested.

Critique 2: o Fig S1: Explain each figure in the legend. The legend currently states "Data characteristics are labelled above graphs in the order of "number of samples in the arbitrary groups (S1,S2), maximum number of taxa in any one sample, and dissimilarity in microbiome composition" Fig S1A has "10,10,0.1" above the graph - Does this mean 10 samples in each group, 10 taxa max in any sample, and 0.1 dissimilarity? Please fully explain each figure A, B, C, etc or more clearly label what the 3 numbers above each graph are as this was not intuitive.

Response: Thank you. We have now added more detailed information on what those numbers mean (Figure S1 legend)

Critique 3: o Line 185 -"and a strong impact on" - should be "HAD a strong impact on"?

Response: Thank you. We have changed it accordingly.

Critique 4: o Why Margalef's species richness as opposed to other alpha diversity metrics? (Would be helpful to explain in manuscript).

Response: Based on critiques above, we have now added both unweighted and weighted diversity metrics, as well as an additional metric of differential abundance analysis. All figures have been updated accordingly and text was revised where needed (with tracked changes). We could not include any phylogenetic measures given that these data are simulated.

Critique 5: o Further explaining the results in biological terms would be helpful. For example: You state that "Sample number had only marginal impacts on the statistical outcomes for alpha and beta diversity" Fig 2a and b show increased sample number with statistically increased $-\log_{10} p$ value. What does this mean? P values get smaller as sample number increases? Is there a threshold sample number under which you cannot obtain significance? (Or if there are low sample numbers that obtain significance, what are the other qualities of this data set that lend themselves to significance?) This type information would help me interpret these data as I consider future studies. The same is true for Fig 2B, 3A/B and 4A/B

Response: Thanks. Yes, as $-\log_{10} p$ -value increases, raw p-values decrease. We have now made this more clear (lines 437-440). We have also analyzed the data further to quantify what thresholds for each metric are the likely barriers to significance based on the regression analyses. These have been updated in the figures.

Critique 6: o For Fig 5 and 6 the y axes are "fold change in p value" - is this fold change increase? Decrease? Either?

Response: We have clarified this in the manuscript (lines 444-446). Specifically, values above 1 are an increase and values below 1 are a decrease.

Critique 7: o It seems unusual that contaminant abundances would not alter alpha diversity outcomes. (Although I understand that what you are capturing is actually p value fold change across 2 simulated groups with 600 datasets.) How should I apply this information? For example, if I am considering UTI (high biomass) vs. healthy (low biomass) urine. The healthy urine is more likely to amplify contaminant sequences. Should I interpret from your results that I don't need to worry about the "extra diversity" from contaminants in the healthy samples as it is ultimately unlikely to change the p value between the UTI/healthy group? (Explaining some of your results in terms of other example studies can help the readers more tangibly grasp the value of this work while also serving as a "gut check" on data interpretations.

Response: Thank you. The non-intuitive results on alpha diversity are driven by the fact that contaminants, in samples processed under the same conditions (location, personnel, instruments, etc), would exhibit a roughly equal distribution across all samples within a single study. We have now also included analyses on the impact of weighted contaminants, as it would be in a UTI vs. healthy case-

control cohort. Interestingly, there was less of an impact of weighted contamination than unweighted. Figures and text have been updated accordingly to reflect this.

Critique 8: o There was a strong impact of contaminant number on the number of differentially abundant taxa. What would be very helpful to know beyond this is how the contaminant numbers shape the differential taxa outcomes. For example, at high contaminant numbers, are the contaminant taxa showing up as differentially abundant (HUGE concern), or are they just pushing "borderline" significant taxa to significance (slightly less concern)? Or are they elevating other random taxa to significance (Concerning)? Or all of the above. Highlighting a few examples within the range of your results will make this much more tangible and applicable for readers.

Response: We have now added a new supplemental figure detailing the distribution of contaminant, borderline, and random taxa that were pushed to significance with contaminations (**Lines 240-243, 448-454 and Figure S2**).

Re: mSystems00408-25R1 (**Impact of study design, contamination, and data characteristics on results and interpretation of microbiome studies**)

Dear Dr. Miller:

Your manuscript has been accepted, and I am forwarding it to the ASM production staff for publication. Your paper will first be checked to make sure all elements meet the technical requirements. ASM staff will contact you if anything needs to be revised before copyediting and production can begin. Otherwise, you will be notified when your proofs are ready to be viewed.

Sincerely,
Jack Gilbert
Editor
mSystems